# Sex Differences in Patent Ductus Arteriosus Incidence and Response to Pharmacological Treatment in Preterm Infants: A Systematic Review, Meta-Analysis and Meta-Regression

**DOI:** 10.3390/jpm12071143

**Published:** 2022-07-14

**Authors:** Moreyba Borges-Lujan, Gema E. Gonzalez-Luis, Tom Roosen, Maurice J. Huizing, Eduardo Villamor

**Affiliations:** 1Department of Neonatology, Complejo Hospitalario Universitario Insular Materno-Infantil (CHUIMI) de Canarias, 35016 Las Palmas de Gran Canaria, Spain; mborluj@gobiernodecanarias.org (M.B.-L.); ggonlui@gobiernodecanarias.org (G.E.G.-L.); 2Department of Pediatrics, Maastricht University Medical Centre (MUMC+), School for Oncology Reproduction (GROW), 6202 Maastricht, The Netherlands; tjcmroosen@gmail.com (T.R.); m.huizing@mumc.nl (M.J.H.)

**Keywords:** sex differences, patent ductus arteriosus, preterm infants

## Abstract

A widely accepted concept in perinatal medicine is that boys are more susceptible than girls to complications of prematurity. However, whether this ‘male disadvantage of prematurity’ also involves persistent patent ductus arteriosus (PDA) has been scarcely investigated. Our aim was to conduct a systematic review and meta-analysis on studies addressing sex differences in the risk of developing PDA among preterm infants. We also investigated whether the response to pharmacological treatment of PDA differs between boys and girls. PubMed/Medline and Embase databases were searched. The random-effects male/female risk ratio (RR) and 95% confidence interval (CI) were calculated. We included 146 studies (357,781 infants). Meta-analysis could not demonstrate sex differences in risk of developing any PDA (37 studies, RR 1.03, 95% CI 0.97 to 1.08), hemodynamically significant PDA (81 studies, RR 1.00, 95% CI 0.97 to 1.02), or in the rate of response to pharmacological treatment (45 studies, RR 1.01, 95% CI 0.98 to 1.04). Subgroup analysis and meta-regression showed that the absence of sex differences was maintained over the years and in different geographic settings. In conclusion, both the incidence of PDA in preterm infants and the response rate to pharmacological treatment of PDA are not different between preterm boys and girls.

## 1. Introduction

The ductus arteriosus (DA) is an essential fetal blood vessel that connects the pulmonary artery to the aorta and serves to shunt blood away from the lungs into the umbilical placental circulation where gas exchange takes place [1,2,3,4,5,6,7]. At birth, closure of the DA is a critical event in the transition to the postnatal circulatory pattern. However, there are situations in which DA closure does not occur or is delayed, resulting in the condition known as persistent patent DA (PDA) [1,2,3,4,5,6,7]. In term and late-preterm infants, PDA is a relatively rare condition that frequently is related to inherent abnormality of the DA and/or signaling pathways that normally trigger its closure [7,8]. In contrast, PDA is very common among very preterm infants because it is generally due to developmental immaturity. PDA would likely not be present if the infant had been born at term [7,8]. The clinical consequences and the therapeutic approach to PDA in very preterm infants are matters of an intense debate that still seems far from being resolved [9,10].

Male–female differences in human health and disease have been recognized for many years [11,12,13,14]. A widely accepted concept in perinatal medicine is the so-called ‘male disadvantage of prematurity’. This concept is supported by a large body of evidence showing that boys are more susceptible than girls to adverse outcomes of prematurity, including bronchopulmonary dysplasia (BPD), retinopathy of prematurity (ROP), necrotizing enterocolitis (NEC), intraventricular hemorrhage (IVH), periventricular leukomalacia (PVL), chronic neurodevelopmental and cognitive impairment, and mortality [15,16,17,18,19]. However, whether PDA is one of the complications of prematurity that is affected by this male disadvantage has not been thoroughly investigated. Interestingly, time to spontaneous DA closure in healthy full-term neonates is longer in girls than in boys [20] and male-to-female ratio for PDA in term infants is about 1:2 in most reports [7,8,21,22,23,24,25]. However, as mentioned above, PDA in term and preterm newborns are two conditions with a different etiopathogenic background and therefore may have a different sex ratio. The results of two recent meta-analyses suggest that there is no sex difference in the risk of developing PDA among preterm infants [19,26], but these meta-analyses were limited by the small number of studies included.

The aim of this systematic review is to answer the question of whether there are sex differences in the risk of developing PDA in preterm infants. We used a broad search strategy in order to include a large number of studies. We also investigated whether the response to pharmacological treatment of PDA differs between boys and girls. Finally, we used subgroup analysis and meta-regression to elucidate whether there are geographic differences or changes over the years in sex differences in PDA risk. 

## 2. Materials and Methods

The methodology of this study is based on that of earlier studies of our group on risk factors for PDA [27,28,29,30]. The study was performed and reported according to the preferred reporting items for systematic reviews and meta-analyses (PRISMA) and meta-analysis of observational studies in epidemiology (MOOSE) guidelines [15]. Review protocol was prospectively registered in the PROSPERO international register of systematic reviews (ID = CRD42018095509). The two research questions were “Do preterm boys have a higher risk of developing PDA than preterm girls?” and “Do preterm boys respond differently than preterm girls to pharmacological treatment of PDA?” 

### 2.1. Sources and Search Strategy

A comprehensive literature search was undertaken using the PubMed/Medline and EMBASE databases. Databases of grey literature were not searched. The search strategy is detailed in Appendix A. No language limit was applied. The literature search was updated up to December 2021. Narrative reviews, systematic reviews, case reports, letters, editorials, and commentaries were excluded, but read to identify potential additional studies. Additional strategies to identify studies included a manual review of reference lists from key articles that fulfilled our eligibility criteria, use of “related articles” feature in PubMed, and use of the “cited by” tool in Web of Science and Google scholar.

### 2.2. Study Selection and Definitions

Studies were included if they had a prospective or retrospective cohort design, examined preterm infants (GA < 37 weeks) and reported primary data that could be used to measure the association between infant sex and (1) rate of PDA and (2) response to pharmacological treatment. Studies that exclusively included late preterm infants (GA ≥ 34 weeks) or combined preterm and term infants were excluded. To identify relevant studies, two reviewers (MB-L, TR) independently screened the results of the searches and applied inclusion criteria using a structured form. Discrepancies were resolved by two other reviewers (GG-L, EV). The studies in English, Spanish, French, German, Dutch, Italian, Portuguese, Catalan, and Galician were directly analyzed by one of the authors with knowledge of the language. Articles in other languages were translated into English using an electronic translator (DeepL^®^, Cologne, Germany). If the translation was unclear, the articles were excluded.

As in our previous meta-analyses [27,28,29,30], the studies were divided according to the way they considered small ductal shunts [27,28,29,30]. Studies comparing small + large PDA vs. closed DA were classified as reporting on “any PDA.” Studies comparing large PDA vs. small PDA + closed DA were classified as reporting on “hemodynamically significant PDA” (hsPDA). Since in our previous meta-analyses we detected several studies that did not consider closed DAs and only compared hsPDA with small PDA, we conducted a meta-analysis in which closed DAs were not taken into account. We assumed the definition “PDA requiring treatment” as a proxy for hsPDA. Regarding response to drug treatment, when a study reported on several treatment courses, only the final response was taken into account. 

### 2.3. Data Extraction and Assessment of Study Quality

Two investigators (MB-L and TR) extracted data on study design, demographics, rate of PDA and/or hsPDA, and response to treatment. A second group of investigators (MJH, GG-L and EV) checked the data extraction for completeness and accuracy. Methodological quality was assessed using the Newcastle-Ottawa Scale (NOS) for cohort studies [15]. This scale assigns a maximum of 9 points (4 for selection, 2 for comparability, and 3 for outcome). NOS scores ≥ 7 were considered high-quality studies (low risk of bias), and scores of 5 to 6 denoted moderate quality (moderate risk of bias) [15].

### 2.4. Statistical Analysis

Studies were combined and analyzed using comprehensive meta-analysis V3.0 software (Biostat Inc., Englewood, NJ, USA). Due to anticipated heterogeneity, summary statistics were calculated with a random-effects model. This model accounts for variability between studies as well as within studies. The risk ratio (RR) with a 95% confidence interval (CI) was calculated. Statistical heterogeneity was assessed by Cochran’s *Q* statistic and by the *I*^2^ statistic. *I*^2^ was interpreted on the basis of Higgins and Thompson criteria, where 25%, 50%, and 75% correspond to low, moderate, and high heterogeneity, respectively [31]. Potential sources of heterogeneity were assessed through subgroup analysis and/or random effects (method of moments) univariate meta-regression analysis as previously described [16,17]. For both categorical and continuous covariates, the R^2^ analog, defined as the total between-study variance explained by the moderator, was calculated based on the meta-regression matrix. Predefined sources of heterogeneity included the following characteristics of cohorts: mean or median GA, median year of birth, geographical location (continent), and drug used for PDA treatment. We used the Egger’s regression test and funnel plots to assess publication bias. Subgroup analyses, meta-regression, and publication bias assessment were performed only when there were at least ten studies in the meta-analysis. A probability value of less than 0.05 (0.10 for heterogeneity) was considered statistically significant.

## 3. Results

### 3.1. Description of Studies and Quality Assessment 

The flow diagram of the search process is shown in Appendix A. Of 2130 potentially relevant studies, 146 (including 357,781 infants) were included [32,33,34,35,36,37,38,39,40,41,42,43,44,45,46,47,48,49,50,51,52,53,54,55,56,57,58,59,60,61,62,63,64,65,66,67,68,69,70,71,72,73,74,75,76,77,78,79,80,81,82,83,84,85,86,87,88,89,90,91,92,93,94,95,96,97,98,99,100,101,102,103,104,105,106,107,108,109,110,111,112,113,114,115,116,117,118,119,120,121,122,123,124,125,126,127,128,129,130,131,132,133,134,135,136,137,138,139,140,141,142,143,144,145,146,147,148,149,150,151,152,153,154,155,156,157,158,159,160,161,162,163,164,165,166,167,168,169,170,171,172,173,174,175,176,177]. Their characteristics are summarized in Appendix A. The quality score of each study according to the Newcastle-Ottawa Scale is depicted in Appendix A. All studies received at least seven points indicating a low risk of bias.

### 3.2. Meta-Analysis

As mentioned in the methods, we carried out four meta-analyses (Table 1): (1) Any PDA (hsPDA + non-hsPDA vs. closed DA) (Figure 1); (2) hsPDA vs. non-hsPDA + closed DA (Figure 2); (3) hsPDA vs. non-hsPDA (Figure 3); and (4) Response to pharmacological treatment (Figure 4). None of the four meta-analyses could demonstrate the presence of significant male-female differences. Heterogeneity was low to moderate in all four meta-analyses (Table 1). Neither visual inspection of funnel plots (Appendix A) nor Egger’s test suggested publication or selection bias for any of the meta-analyses.

### 3.3. Subgroup Analysis and Meta-Regression

As shown in Figure 4, subgroup analysis based on the medication used to treat PDA showed no significant sex differences for any of the drugs. As shown in Table 2, when the studies were divided according to whether they included exclusively extremely preterm infants (i.e., GA < 29 weeks) or also included infants with GA above 28 weeks, the meta-analyses showed no significant sex differences. In addition, subgroup analysis based on the geographic location (continent) of the studies showed no significant male–female differences for any of the continents analyzed (Table 3).

Meta-regression showed that the effect size of the association between male sex and PDA has remained stable over time. That is, it does not correlate with the median year of birth of the cohort (Appendix A). The meta-regression also could not demonstrate a correlation between the mean/median GA of the cohort and the effect size of the association between sex and PDA (Appendix A). 

## 4. Discussion

To our knowledge, this study is the largest and most comprehensive systematic review and meta-analysis on sex differences in PDA. Our results suggest that both the incidence of PDA in preterm infants and the response rate to pharmacological treatment of PDA are not different between preterm boys and girls. Moreover, subgroup analysis and meta-regression showed that the absence of sex differences in PDA is maintained over the years and in different geographic settings.

In a previous meta-analysis, we investigated the male disadvantage for the most important complications of prematurity and found an increased risk of IVH, BPD, ROP, NEC, late onset sepsis, and mortality in preterm boys [19]. However, this was not the case for PDA. In that previous analysis, we included only studies in which infant sex was the independent variable and outcome the dependent variable. This allowed the comparison of the different outcomes but at the cost of including only 18 studies on PDA. A recent meta-analysis by Liu et al. evaluated several risk factors for developing PDA in preterm infants [26]. They found that the risk of developing PDA was slightly higher for boys than for girls but only included 28 studies in the analysis on sex differences. The major strength of the present meta-analysis is the comprehensive database search to identify all the potential studies. Thus, the 146 included studies encompassed a total population of 357,781 infants from 36 different countries and were conducted over more than 35 years. Therefore, our data provide strong evidence for the lack of sex differences in PDA incidence and response to treatment among preterm infants. 

A common conception among neonatologists is the strong interaction between DA closure in the first days of life of preterm infants and their respiratory evolution. The presence of a hemodynamically significant PDA is frequently suspected on the basis of respiratory findings, such as increased oxygen or mechanical ventilation requirements [178,179]. Conversely, changes in pulmonary precapillary tone as consequence of respiratory distress evolution and/or therapy can alter the left-to-right PDA shunt [178,179]. In addition, infants with a moderate to large PDA are at the greatest risk of developing bronchopulmonary dysplasia (BPD) [178,179]. Interestingly, our previous meta-analysis showed that the respiratory course in the first days of life was more complicated in preterm boys than in preterm girls [19]. Male sex was associated with increased risk of RDS, higher rate of intubation at birth, treatment with surfactant, mechanical ventilation, and pneumothorax [19]. However, our present results suggest that the presence of PDA is unlikely to play a role in these sex differences in respiratory courses.

It has been proposed that the male disadvantage begins in utero, when gonadal steroid production already differs strongly by sex [180]. Therefore, the presence of sex differences in DA development and closure might be plausible from a developmental biology perspective. Conventionally, DA closure is divided into two sequential steps: an initial functional closure, mediated by constriction of DA smooth muscle, followed by anatomical vessel remodeling, leading to luminal obliteration [4,5]. Oxygen-mediated contraction and withdrawal of prostaglandin E_2_-induced relaxation contribute synergistically to the initial constriction [4,5]. As mentioned in the Introduction, it has been reported that, among term newborns, the first phase of functional closure of the DA is more rapid in boys than in girls [20], but there is no evidence that this can be extrapolated to the immature DA. Our results do not point in that direction.

To the best of our knowledge, the only model in which it has been studied potential sex differences in the maturation of DA reactivity is the chicken embryo model [181]. Interestingly, chicken embryo development takes place under very different hormonal environments with respect to estrogen levels [182]. In humans, the main source of estrogens is the placenta and, therefore, both female and male fetuses are exposed to high concentrations of the hormone. In the chicken, female embryos have much higher estrogen concentrations than male embryos [182]. Estrogens are also vasoactive in the DA of embryos of both sexes [181]. However, Flisenberg et al. found no differences between males and females in either oxygen-mediated ductal contraction or DA response to different vasoactive agents [181]. Moreover, maturation of the DA contractile response occurred at the same rate in both sexes [181]. Unfortunately, this study has not yet been reproduced in a mammalian model.

Sex differences in pharmacokinetics and pharmacodynamics are common for many drugs and contribute to individual differences in their efficacy and toxicity [183,184,185]. The inhibitors of prostaglandin synthesis indomethacin and ibuprofen are the most frequently used drugs in the treatment of PDA [186]. Paracetamol has been added to these drugs in recent years [186]. Evidence from several experimental and clinical studies indicates differences in levels of prostaglandins and other eicosanoids, as well as in the activity or expression of their synthesizing and metabolizing enzymes between adult males and females [187,188,189]. In addition, there is also substantial evidence of sex differences in the pharmacokinetics and pharmacodynamics of indomethacin, ibuprofen and paracetamol in adults [190,191,192]. Moreover, the preventive effects of indomethacin on IVH in preterm infants appear to be sex-specific [193,194]. In contrast, and although some studies have reported sex differences in the efficacy of the drugs used for pharmacological treatment of PDA [33,49,105], the meta-analysis did not confirm these differences. 

Using subgroup and meta-regression analyses, we examined the influence of geographic and temporal factors on the results of the meta-analysis. Our data suggest that the absence of sex differences in PDA incidence and response to pharmacological treatment can be found in all geographical areas and remains unchanged over the years. In contrast, our previous meta-analysis showed that geographic location and age of cohorts affected sex differences in some of the outcomes of prematurity. Thus, the male disadvantage in periventricular leukomalacia was significantly lower in the cohorts from America when compared with Asian and European cohorts and the male disadvantage in mortality decreased progressively over the years. The analysis of time factor is particularly relevant for PDA because neonatologists have been debating for decades what a hemodynamically significant PDA is, what the health consequences of the presence of a ductal shunt for the preterm newborn are and when and how PDA should be treated [1,9,10]. Our results suggest that these changes as well as advances in neonatology have not affected the sex ratio of preterm PDA. 

Finally, we analyzed the effect of the gestational age of the cohort on the results of the meta-analysis. Since the incidence of term PDA is higher in girls than in boys [7,8,21,22,23,24,25], it would possible that those studies including more mature infants had a different sex ratio than those exclusively focused on extremely preterm infants. However, neither the meta-regression nor the subgroup study confirmed this hypothesis. The mean gestational age of the cohort did not correlate with the sex ratio for PDA. Moreover, subgroup analysis in which only extremely preterm infants (gestational age ≤ 28 weeks) were included showed no sex differences either in the incidence of PDA or in response to pharmacological treatment.

## 5. Conclusions

Although the occurrence of sex differences in incidence and response to treatment of PDA among preterm infants would be plausible from a biological, pharmacological or clinical point of view, the current evidence does not support these differences. Nevertheless, sex is increasingly being recognized as a key variable in the regulation of physiology, pathology, pharmacology and therapeutics, calling for more consideration of sex differences in biomedical research [187,189,195,196,197]. The current clinical approach to PDA therapy in preterm infants is less aggressive with a propensity to observe physiological evolution without pharmacological intervention [1,9,10]. There is a growing body of information on this approach to PDA, and it would be necessary for future studies to take into account the sex variable in their design, sample size calculation, and reporting of the results.

## Figures and Tables

**Figure 1 jpm-12-01143-f001:**
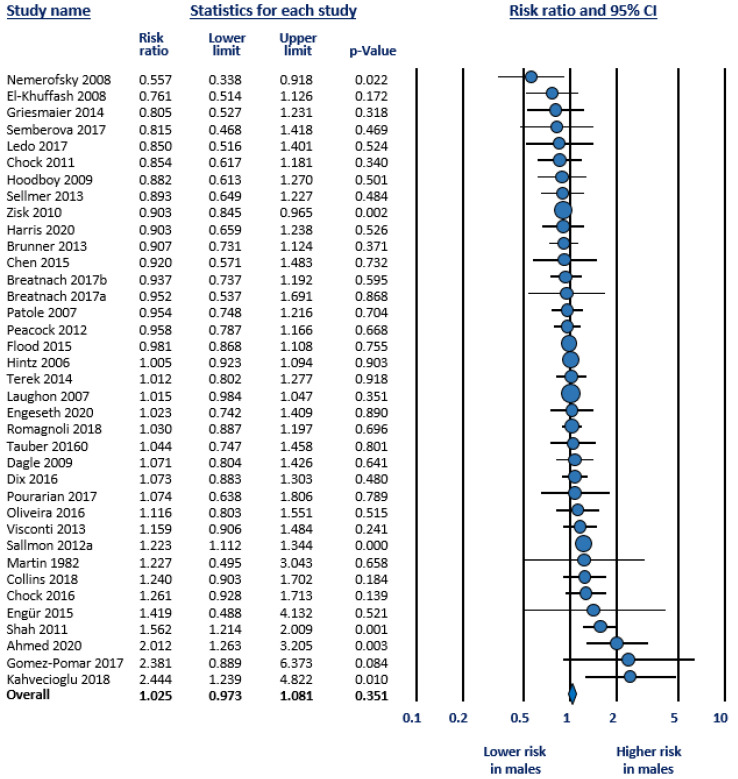
Meta-analysis on the association between sex of preterm newborns and risk of developing any patent ductus arteriosus (PDA). Any ductal shunt was compared with closed DA. The median incidence of PDA in the cohorts was 49.8% (range 12.4 to 82.4%); CI: confidence interval; Risk ratio above 1 means higher risk in males.

**Figure 2 jpm-12-01143-f002:**
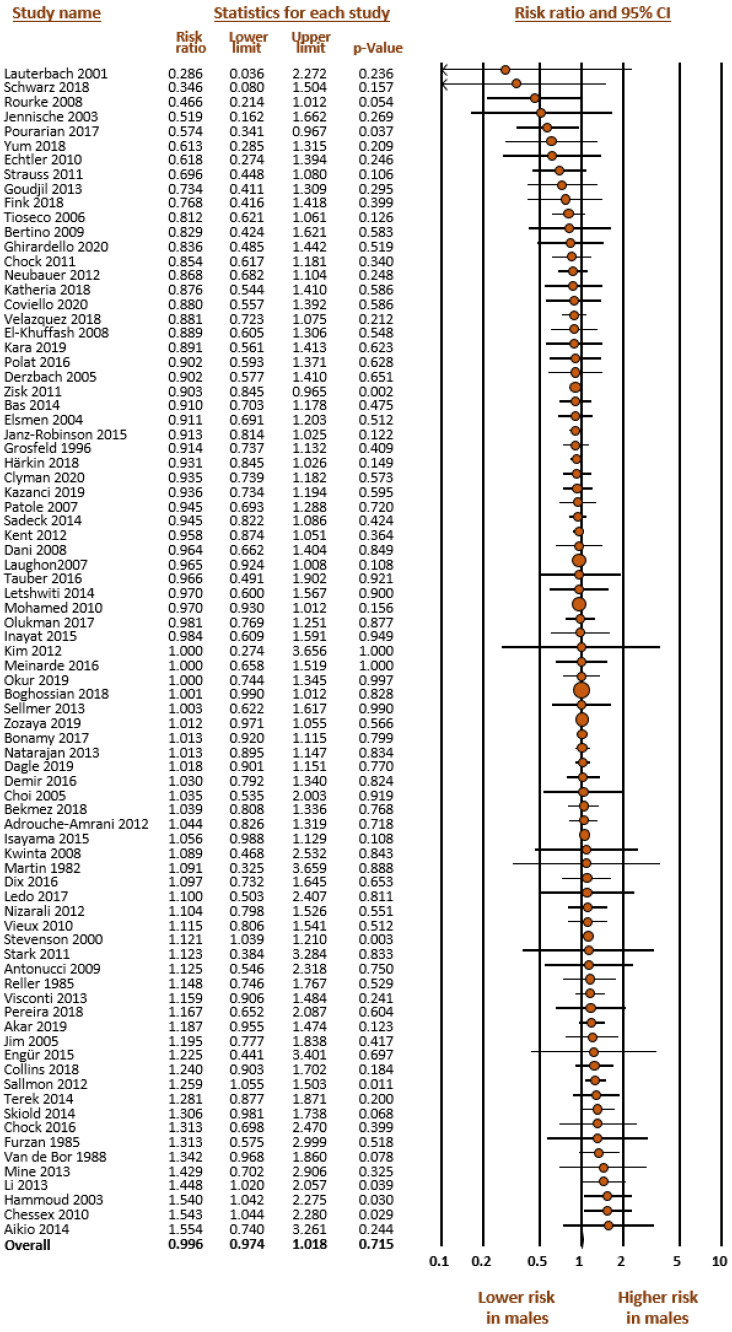
Meta-analysis on the association between sex of preterm newborns and risk of developing hemodynamically significant patent ductus arteriosus (hsPDA). Large ductal shunts were compared with small shunts plus closed DA. The median incidence of hsPDA in the cohorts was 36.7% (range 10.7 to 83.6%); CI: confidence interval; Risk ratio above 1 means higher risk in males.

**Figure 3 jpm-12-01143-f003:**
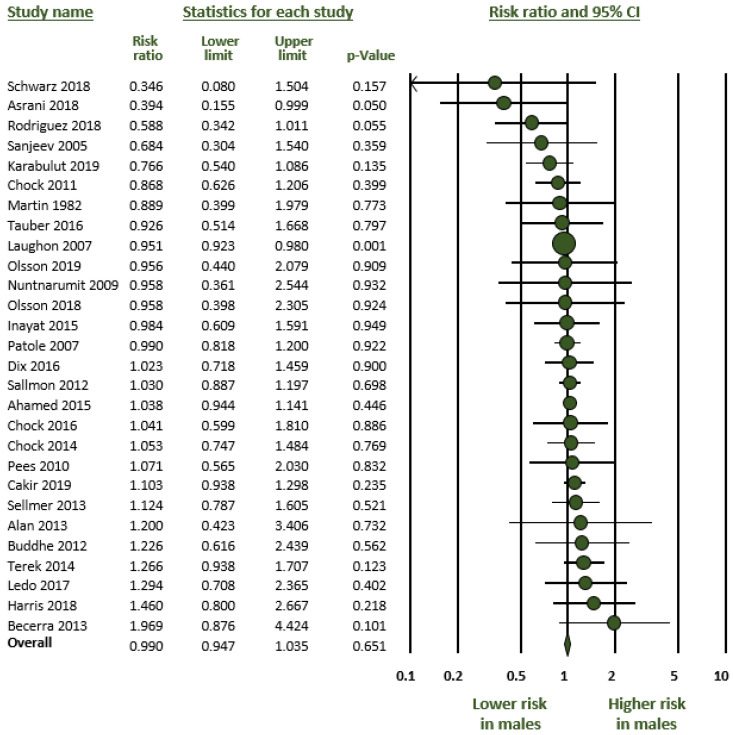
Meta-analysis on the association between sex of preterm newborns and risk of developing hemodynamically significant patent ductus arteriosus (hsPDA). Large ductal shunts were compared with small shunts. The median incidence of hsPDA in the cohorts was 48.3% (range 24.3 to 89.2%). CI: confidence interval; Risk ratio above 1 means higher risk in males.

**Figure 4 jpm-12-01143-f004:**
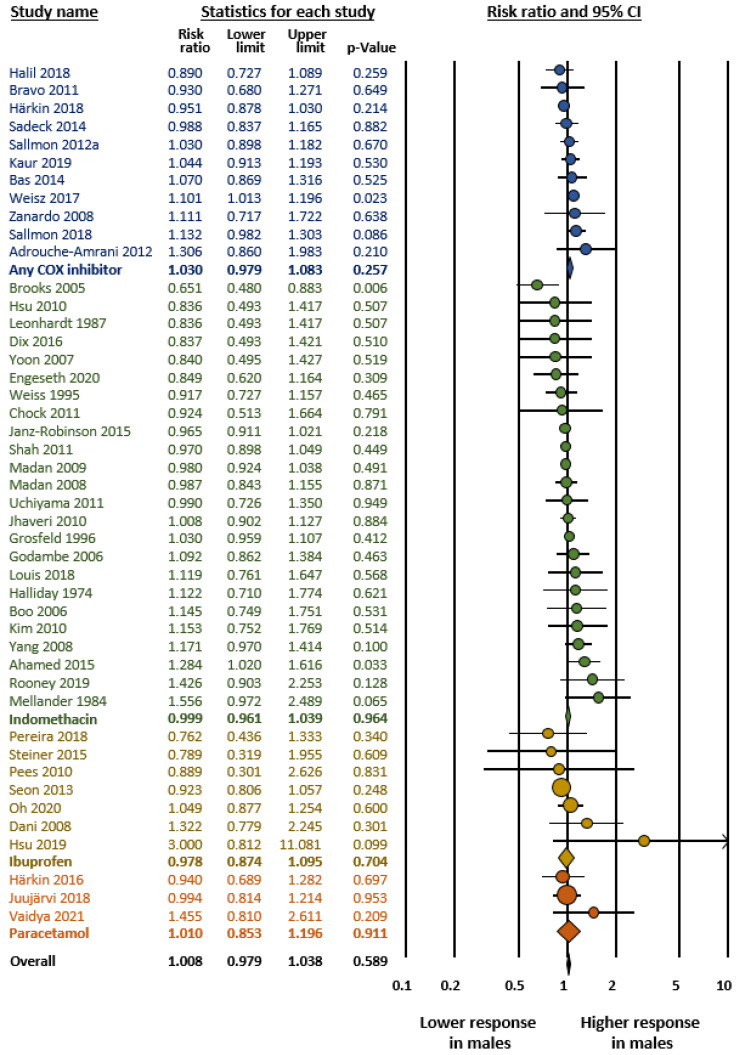
Meta-analysis on the association between sex of preterm newborns and rate of response to pharmacological treatment of patent ductus arteriosus (PDA). CI: confidence interval. COX: cyclooxygenase. Risk ratio above 1 means higher response in males.

**Table 1 jpm-12-01143-t001:** Main meta-analyses.

Meta-Analysis	K	RR	95% CI		*p*	Heterogeneity	
			Lower Limit	Upper Limit		*I*^2^ (%)	*p*
**Any PDA vs. closed DA**	**37**	**1.025**	0.973	1.081	0.351	53.2	0.000
**hsPDA vs. (small PDA + closed DA)**	81	0.996	0.974	1.018	0.715	22.1	0.045
**hsPDA vs. small PDA**	28	0.990	0.947	1.035	0.651	6.0	0.374
**Responders vs. non responders**	45	1.007	0.979	1.036	0.610	14.4	0.207

CI: confidence interval; DA: ductus arteriosus; hs: hemodinamically significant; K: number of studies; PDA: patent ductus arteriosus; RR: risk ratio.

**Table 2 jpm-12-01143-t002:** Subgroup analysis based on inclusion criteria for gestational age.

Meta-Analysis	Subgroup	K	RR	95% CI		*p*	Heterogeneity	
				Lower Limit	Upper Limit		*I*^2^ (%)	*p*
**Any PDA vs. closed DA**	GA < 29 weeks	26	1.023	0.958	1.092	0.502	51.7	0.001
	GA ≥ 29 weeks	10	1.047	0.933	1.175	0.434	53.3	0.023
**hsPDA vs. (small PDA + closed DA)**	GA < 29 weeks	50	1.006	0.979	1.033	0.688	17.7	0.143
	GA ≥ 29 weeks	30	0.977	0.931	1.025	0.334	23.0	0.130
**hsPDA vs. small PDA**	GA < 29 weeks	17	1.024	0.936	1.119	0.609	17.8	0.245
	GA ≥ 29 weeks	11	0.975	0.932	1.019	0.265	0.0	0.690
**Responders vs. non responders**	GA < 29 weeks	40	1.014	0.985	1.043	0.349	18.2	0.160
	GA ≥ 29 weeks	5	0.936	0.854	1.027	0.162	0.0	0.937

CI: confidence interval; DA: ductus arteriosus; GA: gestational age; hs: hemodinamically significant; K: number of studies; PDA: patent ductus arteriosus; RR: risk ratio.

**Table 3 jpm-12-01143-t003:** Subgroup analysis based on continent.

Meta-Analysis	Continent	K	RR	95% CI		*p*	Heterogeneity	
				Lower Limit	Upper Limit		*I*^2^ (%)	*p*
**Any PDA vs. closed DA**	America	14	1.034	0.959	1.116	0.383	64.9	0.000
	Asia	5	1.104	0.892	1.368	0.363	38.8	0.163
	Europe	16	0.992	0.915	1.075	0.841	32.4	0.104
**hsPDA vs. (small PDA + closed DA)**	America	26	0.988	0.958	1.019	0.448	37.5	0.029
	Asia	19	1.021	0.937	1.112	0.640	11.8	0.310
	Europe	30	1.005	0.959	1.052	0.843	7.6	0.348
	Oceania	4	0.940	0.865	1.022	0.149	0.0	0.915
**hsPDA vs. small PDA**	America	12	0.964	0.928	1.001	0.058	3.7	0.409
	Asia	6	1.090	0.958	1.239	0.189	15.3	0.316
	Europe	9	1.012	0.898	1.140	0.849	0.07	0.537
**Responders vs. non responders**	America	18	1.029	0.990	1.069	0.144	17.5	0.245
	Asia	10	1.000	0.921	1.086	0.997	9.8	0.352
	Europe	15	0.996	0.938	1.058	0.905	0.0	0.779

CI: confidence interval; DA: ductus arteriosus; hs: hemodynamically significant; K: number of studies; PDA: patent ductus arteriosus; RR: risk ratio.

## Data Availability

All data relevant to the study are included in the article or uploaded as Appendix A. Additional data are available upon reasonable request.

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
