# Peer review of "Sex Differences in Patent Ductus Arteriosus Incidence and Response to Pharmacological Treatment in Preterm Infants: A Systematic Review, Meta-Analysis and Meta-Regression"

_jpm, 2022, doi:10.3390/jpm12071143_

Round 1

Reviewer 1 Report

Title: Sex differences in PDA incidence and response to pharmacological treatment in preterm infants: A systematic review, meta-analysis and meta-regression.

Summary:

This meta-analysis examined whether there are sex differences in the risk of developing PDA in preterm infants and also investigated whether response to pharmacological treatment of PDA differs in males when compared to females. This study included 146 studies (n=35,7781) infants. The authors concluded that incidence of PDA in preterm infants and response rate to pharmacological treatment of PDA did not differ between preterm males and females.

Abstract: Well written abstract which conveys the reader a good synopsis and entices the reader to download the article to read further.

Introduction: Well written introduction highlighting a gap in knowledge. The introduction culminates with a clear question. Perhaps the authors could consider rephrasing the research question using the PICO format.

Materials and methods:

1.       Appropriate reporting guidelines used and study was registered with PROSPERO. Was the study registered with PROSPERO before carrying out the search?

2.       Line 76: for uniformity with the abstract please use PubMed/Medline

3.       Was grey literature, theses data included?

4.       What did the authors use to navigate around different languages

Results, Discussion and Conclusion: Very well written.

Author Response

  1. Perhaps the authors could consider rephrasing the research question using the PICO format.

Thank you very much for the suggestion. We usually use the PICO or PECO (Population, exposure, control, outcome) format in our meta-analyses. However, since in this meta-analysis we are dealing with sex differences we have chosen to use a research question because sex is a biological characteristic and not an exposure or an intervention.

  1. Appropriate reporting guidelines used and study was registered with PROSPERO. Was the study registered with PROSPERO before carrying out the search?

The study was prospectively registered in PROSPERO. This information is now specified in the manuscript (line 69).

  1. Line 76: for uniformity with the abstract please use PubMed/Medline.

Changed

  1. Was grey literature, theses data included?

Databases of grey literature were not searched. This information is now specified in the manuscript (line 77).

  1. What did the authors use to navigate around different languages?

The studies in English, Spanish, French, German, Dutch, Italian, Portuguese, Catalan, and Galician were directly analyzed by one of the authors with knowledge of the language. Articles in other languages were translated into English using an electronic translator (DeepL). If the translation was unclear, the articles were excluded. This information is now specified in the manuscript (lines 94-98).

Reviewer 2 Report

Congratulations on your work. Maybe it would be nice to show in the introduction the incidence of the PDA

Author Response

Thank you very much for the positive evaluation of our work and for the suggestion. As commented in the article, the diagnostic criteria for PDA have been changing over the years and there is no real consensus. That is why it is very difficult to give data on the incidence of the problem. To adequately reflect this situation, we have included the median incidence and range of incidence of PDA from the different cohorts included in the meta-analysis. This information is now included in the figure legends.